# Rhytidhylides A and B, Two New Phthalide Derivatives from the Endophytic Fungus *Rhytidhysteron* sp. BZM-9

**DOI:** 10.3390/molecules26206092

**Published:** 2021-10-09

**Authors:** Sha Zhang, Dekun Chen, Min Kuang, Weiwei Peng, Yan Chen, Jianbing Tan, Fenghua Kang, Kangping Xu, Zhenxing Zou

**Affiliations:** 1Xiangya School of Pharmaceutical Sciences, Central South University, Changsha 410013, China; ad8879789@163.com (S.Z.); 197211019@csu.edu.cn (D.C.); kuangmin@csu.edu.cn (M.K.); pww199802@163.com (W.P.); chenyanya234@163.com (Y.C.); tanjb1009@csu.edu.cn (J.T.); kangfenghua@csu.edu.cn (F.K.); xukp395@csu.edu.cn (K.X.); 2Hunan Key Laboratory of Diagnostic and Therapeutic Drug Research for Chronic Diseases, Changsha 410013, China

**Keywords:** *Leptospermum brachyandrum*, *Rhytidhysteron* sp. BZM-9, endophyte, phthalate derivative

## Abstract

Two new phthalide derivatives, rhytidhylides A (**1**) and B (**2**), together with ten known compounds (**3**–**12**) were isolated from cultures of *Rhytidhysteron* sp. BZM-9, an endophyte isolated from the leaves of *Leptospermum brachyandrum*. Their structures were identified by an extensive analysis of NMR, HRESIMS, ECD, and through comparison with data reported in the literature. In addition, the cytotoxic activities against two human hepatoma cell lines (HepG2 and SMMC7721) and antibacterial activities against MRSA and *E. coli* were evaluated.

## 1. Introduction

Endophytic fungi plays a role not only in supplying plants with the basic nutrients indispensable for their growth and helping them in the mechanisms of adaptation to various environmental stresses (i.e., salinity, drought), but they can also produce various bioactive natural products [1]. Phthalide, widely found in several higher and lower plant and fungal genera, is a naturally occurring benzobutyrolactones [2]. Many of the naturally occurring phthalides display different biological activities including antibacterial [3], antifungal [4], insecticidal [5], cytotoxic [6], and antioxidant [7] effects, which has also attracted widespread attention from other researchers.

During our search for new bioactive, secondary metabolites of fungi isolated from various medicinal plants, the strain *Rhytidhysteron* sp. BZM-9 was obtained and identified, which was isolated from *Leptospermum brachyandrum*. *Rhytidhysteron* sp. is a clinically pathogenic fungus [8]. The genus *Rhytidhysteron* includes two species: *R. rufulum* and *R. hysterinum*, which has a worldwide distribution and occurs particularly in the tropics and subtropics [8]. As far as we know, only a few chromones and Spirobisnaphthalenes have been reported from the genus *Rhytidhysteron* [9,10,11,12]. In our previous work, some chlorinated cyclopentene and isocoumarin derivatives from this strain were reported [13,14]. In the course of our continued search for bioactive compounds from endophytes, two new phthalide derivatives, rhytidhylides A (**1**) and B (**2**), along with ten known compounds (**3**–**12**), were isolated from cultures of *Rhytidhysteron* sp. BZM-9 (Figure 1). Herein, we describe the isolation, structure elucidation, and biological activities of these compounds.

## 2. Results and Discussion

Compound **1** was obtained as a yellow solid, and the molecular formula was deduced as C_12_H_14_O_5_ based on HRESIMS ion peak at *m/z* 239.0913 [M + H] ^+^ (calcd. for C_12_H_15_O_5_, 239.0914 [M + H] ^+^), indicating six degrees of unsaturation. The ^1^H and ^13^C NMR data (Table 1) of **1** revealed the presence of one ester carbonyl [(*δ*_C_ 171.6 (C-1)], one five substituted benzene ring [*δ*_H_ 6.41 (H-4); *δ*_C_ 99.9 (C-4), 163.3 (C-5), 111.2 (C-6), 155.4 (C-7), 151.6 (C-3a), 102.8 (C-7a)], one oxymethine [*δ*_H_ 3.87 (q, *J* = 6.5 Hz, H-8); *δ*_C_ 71.1 (C-8)], one oxygenated sp^3^ non-protonated carbon [(*δ*_C_ 89.5 (C-3)], and three methyls [*δ*_H_ 2.06 (H-11), 1.57 (H-10), and 1.08 (d, *J* = 6.5 Hz, H-9); *δ*_C_ 6.4 (C-11), 20.7 (C-10), 16.1 (C-9)]. Comparison of the ^1^H and ^13^C NMR data (Table 1) of **1** with those of known compound **3** suggested that both compounds shared similar structural features, with the difference being that one proton of C-8 was replaced by a hydroxyl group. The HSQC correlation of C-8 with a methine proton (*δ*_H_ 3.87) rather than a methylene proton and obvious down-field chemical shift of C-8 (*δ*_C_ 71.1) verified this conclusion. Thus, the planar structure of **1** was completely established, which was further supported by HMBC and ^1^H-^1^H COSY correlations, as present in Figure 2.

Compound **1** has two asymmetric centers. The NOESY correlations were not conclusive in the case of **1**. Thus, to determine the relative configuration of **1**, gauge-independent atomic orbital (GIAO) DFT ^13^C NMR calculations were performed at the ɷB97x-D/6-31G* level using MeOH as the solvent, and the calculations data were compared with their experimental values, following the reported STS protocol. According to linear regression analysis of ^13^C NMR chemical shifts, the values of the correlation coefficient (*R*^2^) were 0.9983 for **1a** and 0.9987 for **1b** (Figure 3. Moreover, the resulting *P_mean_* and *P_rel_* parameters as well as MAE and RMS values further showed **1b** or its enantiomers are correct structures for **1** (Table 2). Subsequently, the absolute configurations of **1** were determined to be 3*R*,8*S* on the basis that the experimental ECD perfectly matched with the calculated ECD (Figure 4). Therefore, compound **1** was named rhytidhylide A.

Compound **2** was obtained as a pale yellow solid with the molecular formula of C_13_H_16_O_5_, which was deduced from the positive HRESIMS ion as *m/z* 253.1073 [M + H] ^+^ (calcd for C_13_H_17_O_5_, 253.1076 [M + H] ^+^), implying six degrees of unsaturation. Comparison of the 1D and 2D NMR data (Table 1) of **2** and **1** suggested that they shared closely similar NMR resonances. The main difference between them was the exhibition of a methoxy group at the C-7 position of **1** and a hydroxyl group C-7 position of **1**. This conclusion could be further verified by the HMBC correlation from -OCH_3_ (*δ*_H_ 3.86) to C-7 (*δ*_C_ 157.5) (Figure 2). Hence, the planar structure of **2** was corroborated.

Similarly, the relative configuration of **2** was assigned by (GIAO) DFT ^13^C NMR calculations. By comparing the calculated ^13^C NMR data with the corresponding experimental values, the possible configuration of **2** was distinguished (Figure 3) (Table 2). Finally, the absolute configuration was established as 3*S*, 8*R* based on the experimental ECD, which was highly similar to the calculated ECD (Figure 4). Thus, compound **2** was named rhytidhylide B.

In addition to the two new phthalide derivatives, rhytidhylides A and B, ten known compounds were isolated and identified as 3-ethyl-5,7-dihydroxy-3,6-dimethylphthalide (**3**) [15], 5-hydroxy-7-methoxy-4,6-dimethylphthalide (**4**) [16], 4-hydroxy-6-methoxy-7-methyl-3-oxo-1,3-dihy-dro-isobenzofuran-5-carbaldehyde (**5**) [17], 4-hydroxy-6-methoxy-5-methyl-1(3H)-isobenzofuranone (**6**) [18], altechromones A (**7**) [19], 2-methyl-5-methylcarboxymethyl-7-hydroxychromone (**8**) [20], 2-(2’-hydroxypropyl)-5-methyl-7-hydroxychromone (**9**) [21], 4-O-methylsclerone (**10**) [22], 1H-indole-3-carboxaldehyde (**11**) [23], and euphorbol (**12**) [24], by comparing their experimental spectral data with the reported spectral data in the literature. In addition, the crystallographic data of compound **4** were reported for the first time (Figure 5).

A review has reported that phthalide derivatives are present in nature with an enormous spectrum of bioactivities extending from bactericidal to cytotoxic [2]. Additionally, methicillin-resistant *Staphylococcus aureus* (MRSA) is endemic in hospitals worldwide and causes substantial morbidity and mortality, which is becoming an important public health problem [25]. Therefore, all compounds were evaluated for their cytotoxic activities inhibited by two human hepatoma cell lines (HepG2 and SMMC7721) and antibacterial activities against MRSA and *E. coli*. Among them, compound **12** displayed weak antibacterial activity against MRSA with a MIC value of 62.5 ug/mL (Table 3) (Appendix A), while none of the compounds showed cytotoxicities at the tested condition (IC_50_ > 80 µM) (Appendix A).

## 3. Materials and Methods

### 3.1. General Experimental Procedures

Optical rotations were measured on a Rudolph Research Analytical Autopol IV automatic polarimeter. HRESIMS spectra were recorded on an Agilent 6500 series Q-TOF mass spectrometer (Agilent, Singapore) analyser with a positive ion mode. Experimental ECD spectra were performed on a chirascan plus Circular Dichroism spectrometer. NMR spectra (1D and 2D) were obtained using Bruker 500MHz spectrometers and using TMS as an internal standard. The single crystal data were collected on an Agilent Xcalibur Novasingle-crystal diffractometer equipped with CuKα radiation. Preparative HPLC was done using an Agilent 1100 prep-HPLC system with a YMC-peak ODS-A column (5 µm, 250 × 10 mm). Sephadex LH-20 (GE Healthcare, Uppsala, Sweden), silica gel (200–300 and 60–100 mesh, Qingdao Marine Chemical Factory, Qingdao, China), and C_18_ reversed-phase silica gel (40–75 µm, Fuji, Kasugai, Japan) were used for column chromatography (CC). All solvents were of analytical grade.

### 3.2. Fungal Material

The fungal strain *Rhytidhysteron* sp. BZM-9 was isolated from the leaves of *Leptospermum brachyandrum*, collected in the South China Botanical Garden in Guangzhou city, Guangdong province of China, in September 2016. The strain was deposited at the Xiangya School of Pharmaceutical Sciences, Central South University. The fungus was identified as *Rhytidhysteron* sp. BZM-9 (GenBank accession number: MN788611).

### 3.3. Fermentation, Extraction and Isolation of Compounds

The fungus was inoculated on solid rice medium, which was prepared by autoclaving 250 g of rice in 300 mL of demineralized water in a 500 mL Erlenmeyer flask. The fermentation was performed in 15 flasks under static conditions at room temperature for a month [26]. The fungal culture was extracted with ethyl acetate added to each flask, and the extract was subsequently dried under a vacuum to afford 50 g.

The crude extract was subjected to silica gel column chromatography eluting with petroleum ether (PE)–ethyl acetate (EtOAc)–methanol (MeOH) (*v/v/v*, 100:0:0–0:0:100) to give ten fractions (Fr. 1 to Fr. 10). Fr. 6 (8.2 g) was further purified by silica gel CC with a gradient of CH_2_Cl_2_-MeOH (*v/v*, 100:0~0:100) to provide eight fractions (Fr. 6-1 to Fr. 6-8). Fr. 6-3 (0.83 g) was further purified by semi-preparative HPLC with MeCN-H_2_O (0~20 min, 25~25%, 3 mL/min) to obtain compound **7** (10.3 mg, *t*_R_ = 17 min), **8** (3.5 mg, *t*_R_ = 14 min), and **9** (5.8 mg, *t*_R_ = 11 min). Fr. 4 was separated by column chromatography over silica gel eluted with PE-EtOAc (*v/v*, 30:1–0:100) and then purified by prep-HPLC with MeCN-H_2_O (0~30 min, 80~95%, 3 mL/min) to give **12** (3.4 mg, *t*_R_ = 16 min).

Fr. 6-4 (1.2 g) was performed on Sephadex LH-20 column with MeOH to get four subfractions (Fr. 6-4-1 to Fr. 6-4-4). Fr. 6-4-1 (0.19 mg) was then purified by semi-preparative HPLC with ACN/H_2_O (0~50 min, 10~30%) to obtain compounds **1** (3.5 mg, *t*_R_ = 26.3 min) and **2** (2.3 mg, *t*_R_ = 29.5 min). Fr. 6-4-2 (0.53 g) was purified by silica gel CC and eluted with a gradient of PE-EtOAc (*v/v*, 30:1~0:100) to provide four fractions (Fr. 6-4-2-1 to Fr. 6-4-2-4). Fr. 6-4-2-2 (95 mg) was further purified by semi-preparative HPLC using MeCN-H_2_O (0~40 min, 30~40%, 3 mL/min) to obtain compounds **3** (7.3 mg, *t*_R_ = 26 min), **4** (2 mg, *t*_R_ = 13.5 min), and **5** (6.6 mg, *t*_R_ = 18.4 min). Fr. 6-4-2-2 (0.31 g) was purified by silica gel column chromatography using a mixture system of PE-EtOAc (*v/v*, 20:1–0:100) to obtain compound **6** (1.6 mg).

Fr. 6-5 (2.3 g) was subjected to a reversed-phase ODS column with a gradient of MeOH-H_2_O (*v/v*, 20:80~100:0) to obtain eight subfractions (Fr. 5-1 to Fr. 5-8). Further purification of fraction Fr. 5-2 (0.21 g) by semi-preparative HPLC (mobile phase: 18% MeCN/H_2_O) yielded compound **11** (2.1 mg, *t*_R_ = 30 min). Fr. 5-3 (0.95 g) was fractionated into three subfractions (Fr. 5-3-1 to Fr. 5-3-3) by silica gel chromatography (CH_2_Cl_2_-MeOH, *v/v*, 100:0~0:100). Fr. 5-3-1 was further purified by semi-preparative HPLC using MeCN-H_2_O (0~30 min, 20~30%, 3 mL/min) to obtain compound **10** (5.4 mg, *t*_R_ = 15 min).

*Rhytidhylide A (**1**)*. Yellow solid; ${\left[\alpha \right]}_{D}^{25}$ −11.6 (*c* 0.18, methanol); HPLC-UV (ACN-H_2_O) *λ*_max_: 226, 262, 295 nm; HRESIMS *m/z* 239.0913 [M + H] ^+^ (calcd for C_12_H_15_O_5_, 239.0914 [M + H] ^+^); ^1^H (500 MHz) and ^13^C (125 MHz) NMR spectral data, see Table 1.

*Rhytidhylide B (**2**)*. Pale yellow solid; ${\left[\alpha \right]}_{D}^{25}$ +5.0 (*c* 0.10, methanol); HPLC-UV (ACN-H_2_O) *λ*_max_: 220, 262, 295 nm; HRESIMS *m/z* 253.1073 [M + H] ^+^ (calcd for C_13_H_17_O_5_, 253.1076 [M + H] ^+^); ^1^H (500 MHz) and ^13^C (125 MHz) NMR spectral data, see Table 1.

### 3.4. X-ray Crystallographic Analysis

Crystal data for **4**. (No. CCDC 2104595) C_11_H_12_O_4_ (*M* = 208.21 g/mol, orthorhombic, space group Pna2_1_ (no. 33), *a* = 7.6421(3) Å, *b* = 16.5289(5) Å, *c* = 7.7031(2) Å, *V* = 973.02(5) Å^3^, *Z* = 4, *T* = 99.9(6) K, *μ*(CuK*α*) = 0.910 mm^−1^, *Dcalc* = 1.421 g/cm^3^, 3840 reflections measured (10.704° ≤ 2Θ ≤ 147.408°), 1671 unique (*R*_int_ = 0.0541, *R*_sigma_ = 0.0359) which were used in all calculations. The final *R*_1_ was 0.0566 (I > 2*σ*(I)) and *wR*_2_ was 0.1561 (all data). Flack parameter = −0.2(3).

### 3.5. Quantum Chemistry Calculations

The conformation optimization, ECD spectrum calculation, and DFT GIAO ^13^C NMR calculation were performed as previously described [14].

### 3.6. Cytotoxic Activity Assay

Cytotoxicities of all compounds were tested against two human hepatoma cell lines (HepG2 and SMMC7721) using a microplate 3-(4,5-dimethylth-iazole-2-yl)-2,5-diphenyltetrazolium bromide (MTT) assay as described previously [13]. Doxorubicin was used as a positive control, and experiments were repeated three times.

### 3.7. Antimicrobial Activity Assay

Antimicrobial activities of all compounds were evaluated by calculating MIC values against MRSA and *E. coli* using the broth microdilution method according to CLSI guidelines [27]. Vancomycin (MIC = 1.25 µg/mL) was used as a positive control.

## 4. Conclusions

In the course of our continued exploration of the fungal strain *Rhytidhysteron* sp. BZM-9 for biologically active metabolites, two new phthalide derivatives, rhytidhylides A (**1**) and B (**2**), together with ten known compounds (**3**-), were isolated and identified. The cytotoxic activities and antimicrobial activities of **1**–**12** were also evaluated, but only compound **12** showed weak inhibitory activity against MRSA. All in all, the activities of these compounds were not thoroughly investigated, just the evaluation of the cytotoxicity and antibacterial assays, and only a few cells and strains have been tested. Other activities and their mechanisms are still worthy of further exploration.

## Figures and Tables

**Figure 1 molecules-26-06092-f001:**
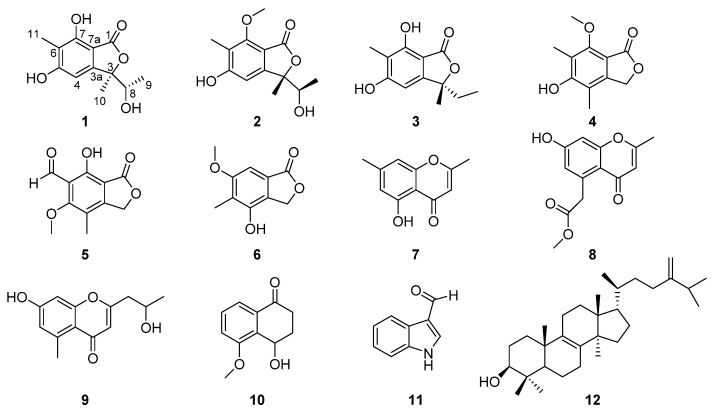
Structures of compounds **1–12**.

**Figure 2 molecules-26-06092-f002:**
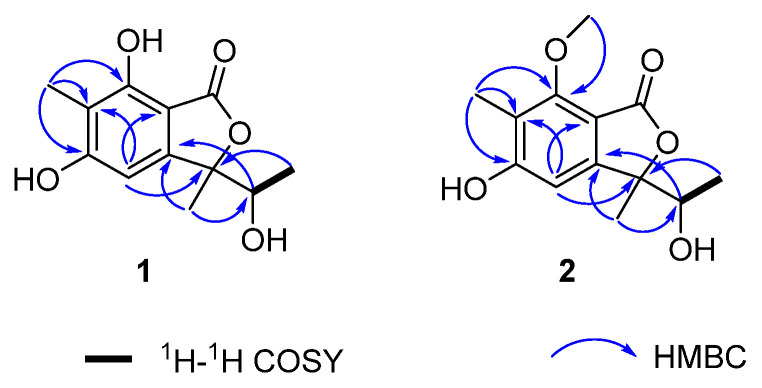
^1^H-^1^H COSY and key HMBC correlations of **1** and **2**.

**Figure 3 molecules-26-06092-f003:**
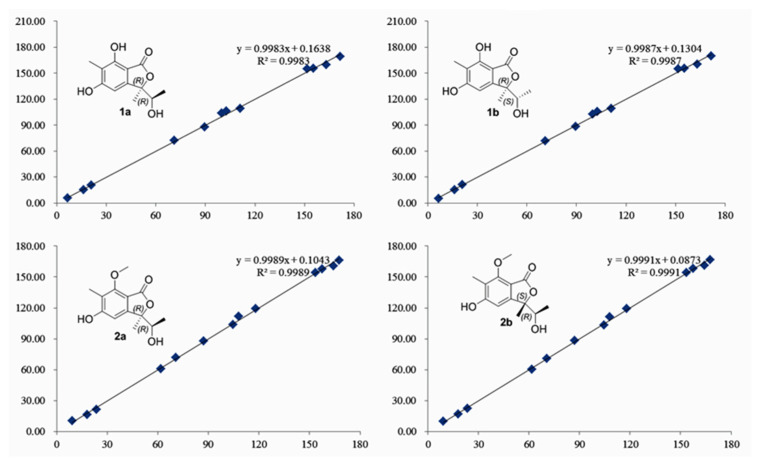
Regression analysis of experimental and calculated ^13^C NMR chemical shifts for **1** and **2**.

**Figure 4 molecules-26-06092-f004:**
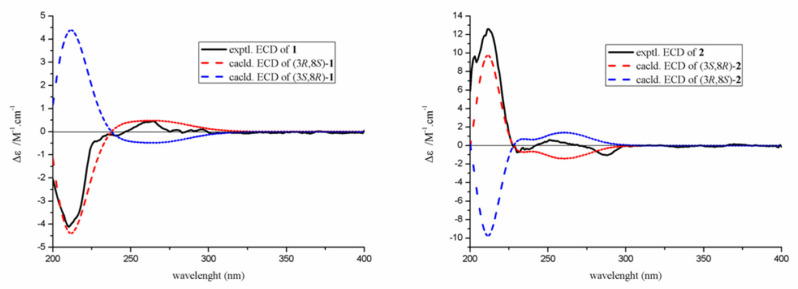
Experimental and calculated ECD spectra of compounds **1** and **2**.

**Figure 5 molecules-26-06092-f005:**
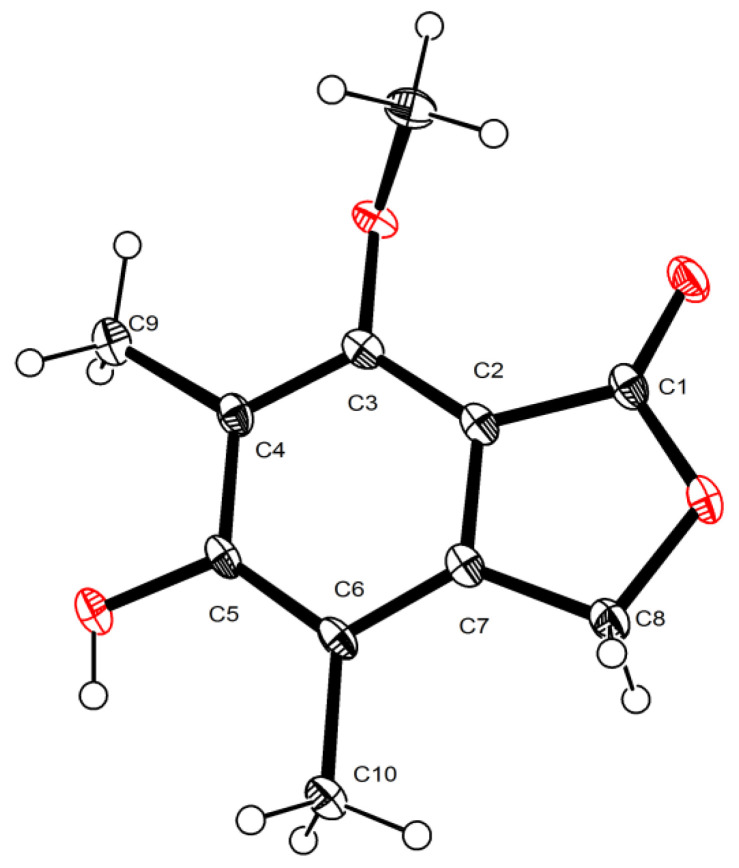
ORTEP drawing of the X-ray structures of **4**.

**Table 1 molecules-26-06092-t001:** ^1^H (500 MHz) and ^13^C NMR (125 MHz) spectral data of **1** and **2** (*δ* in ppm, *J* in Hz).

Position	1 (CD_3_OD)		2 (DMSO-*d*_6_)	
	*δ* _H_	*δ* _C_	*δ* _H_	*δ* _C_
1		171.6		167.6
3		89.5		87.2
4	6.41 (1H, s)	99.9	6.69 (1H, s)	104.5
5		163.3		164.1
6		111.2		118.0
7		155.4		157.5
8	3.87 (1H, q, 6.5)	71.1	3.84 (1H, q, 6.0)	70.5
9	1.08 (3H, d, 6.5)	16.1	0.90 (3H, d, 6.0)	17.9
10	1.57 (3H, s)	20.7	1.47 (3H, s)	23.6
11	2.06 (3H, s)	6.4	2.00 (3H, s)	9.0
3a		151.6		153.6
7a		102.8		107.8
-OCH_3_			3.86 (3H, s)	61.7

**Table 2 molecules-26-06092-t002:** Calculated ^13^C chemical shifts fitting with the experimental data of compounds **1** and **2** following STS protocol.

Exptl.	1				Exptl.	2			
	1a	dev	1b	dev		2a	dev	2b	dev
171.5	169.08	2.42	169.41	2.09	167.6	166.27	1.33	166.58	1.02
89.3	87.98	1.32	88.47	0.83	87.2	87.86	0.66	88.21	1.01
100.4	103.67	3.27	102.82	2.42	104.5	103.97	0.53	103.25	1.25
163.3	159.66	3.64	159.94	3.36	164.1	160.75	3.35	161.08	3.02
111.1	108.95	2.15	108.95	2.15	118	119.44	1.44	119.51	1.51
155.1	155.49	0.39	155.69	0.59	157.5	157.92	0.42	158.15	0.65
70.9	72.32	1.42	71.78	0.88	70.5	71.59	1.09	71.06	0.56
16.1	15.42	0.68	15.64	0.46	17.9	16.63	1.27	16.75	1.15
21.6	20.84	0.76	21.63	0.03	23.6	21.54	2.06	22.41	1.19
6.4	5.80	0.60	5.39	1.01	9	10.31	1.31	9.99	0.99
151.2	154.90	3.70	154.78	3.58	153.6	154.37	0.77	154.10	0.50
103.2	105.98	2.78	105.61	2.41	107.8	111.73	3.93	111.45	3.65
					61.7	60.63	1.07	60.46	1.24
	MAE ^a^	1.93	MAE ^a^	1.65		MAE ^a^	1.48	MAE ^a^	1.37
	RMS ^b^	2.26	RMS ^b^	2.00		RMS ^b^	1.79	RMS ^b^	1.63
	*P_mean_*	19.63%	*P_mean_*	27.76%		*P_mean_*	26.40%	*P_mean_*	32.40%
	*P_rel_*	1.54%	*P_rel_*	98.46%		*P_rel_*	6.51%	*P_rel_*	93.49%

^a^ mean absolute error; ^b^ root mean square.

**Table 3 molecules-26-06092-t003:** Antimicrobial activities of compounds **1–12**.

Compounds	MIC (ug/mL)	
	MRSA	*E. coli*
**1–2, 5, 7**	>500	>500
**3**	250	>500
**4**	250	>500
**6**	125	>500
**8**	125	>500
**9**	125	>500
**10**	125	>500
**11**	500	>500
**12**	62.5	>500
Vancomycin ^a^	1.25	≥40

^a^ positive control.

## Data Availability

All data are available in this publication and in the Appendix A.

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
