# Peer review of "Rhytidhylides A and B, Two New Phthalide Derivatives from the Endophytic Fungus Rhytidhysteron sp. BZM-9"

_molecules, 2021, doi:10.3390/molecules26206092_

Round 1

Reviewer 1 Report

This paper is not yet ready to be published and before it must pass through major changes that may improve this manuscript.

Below the authors will find some comments that will help to improve the article.

The article, in the present format, should be considered as a short communication.

Introduction

The introduction is too short. For example, there is no information about Rhytidhysteron sp. What is its importance? What is it used for? Etc.

The authors added results in the introduction (Figure 1).

Materials and Methods

Please provide references for each methodology described.

Results

Please, add a Figure regarding the antimicrobial activity assay results.

Please, also add a Figure regarding the cytotoxic activity assay results.

Tables and Figures

Use the full scientific name. In addition, use more complete captions, so that readers can understand the table and/or figure without the need to check the text.

The presentation and resolution of figures should be improved.

Reviewer 2 Report

This is an excellent work that could contribute in an important way, when it is finished exploring the possible biological activities that these two new compounds could have, as mentioned in the conclusions of the manuscript, although the characterization of the two new compounds is well documented, the fact that there is no relevant activity in the two types of trials that were carried out,  makes the manuscript a short communication, rather than a complete article, I suggest it be published once you have results of the other trials you are doing.

Author Response

Thanks for the reviewer's suggestion. Phthalide derivatives are present in nature with an enormous spectrum of bioactivities, including antibacterial, antifungal, insecticidal, cytotoxic, and antioxidant effects. Compounds 1 and 2 evaluated for their antibacterial and cytotoxic activities in our research, but neither showed inhibitory activities. However, the amount of compound 1 and 2 isolated from Rhytidhysteron sp. BZM-9 was very little, and the separated amount has been used up by the activities test.

Reviewer 3 Report

This manuscript describes two new phthalide derivatives (Rhytidhylides A and B) from the endophytic fungus Rhytidhysteron sp. BMZ-9. The manuscript was well written and the methods for experiment are classic and sound. The content bears the criteria for publication in this journal. However, there are some issues to be revised before the publication.

1) In Figure 1, authors are suggested to add the information of C-3a, -7a, -11 in the structure of compound 1. And the structure of 4 has wrong angle of OH group.

2) lines 45-47, there are some mistakes; c-7a -> C-7a; one oxymethine is C-8, not C-3. Please correct it. Can you re-check the assignment of C-10 and C-11?

3) In ECD calculation (in Figure 4), authors are suggested to add the calculated ECD curves of enantiomers of 1b and 2b to be more comparable.

4) Authors should correct all ug/ml to ug/mL.

5) Authors can add the information for readers about what MRSA is and why antibacterial activity against MRSA is important.

6) line 138, what is the PE-EtOAc-MeOH? Please add the full names of the solvents.

7) In conclusion, number of the compounds should be bold.

Round 2

Reviewer 1 Report

Materials and Methods

The authors have not yet provided references for each methodology described. For example, see topic 3.3. There is no reference on the methodology of fungal culture, nor for chromatography.

Results

The authors just added the Figure referring to the results of the antimicrobial activity assay in the answer file.

Regarding the Figure of the cytotoxic activity assay results, the authors did not even provide it in the response file

Author Response

Materials and Methods

  1. The authors have not yet provided references for each methodology described. For example, see topic 3.3. There is no reference on the methodology of fungal culture, nor for chromatography.

Reply: Thank you for your kind reminder. The reference of fungal culture was added in our manuscript (reference [26]). However, the chromatography methodology was not based on the reported literature but was explored through TLC and HPLC during the experiment.

Results

  1. The authors just added the Figure referring to the results of the antimicrobial activity assay in the answer file.

Reply: Thanks for the reviewer's suggestion, and antimicrobial activity assay results have been added in Supplementary Material.

  1. Regarding the Figure of the cytotoxic activity assay results, the authors did not even provide it in the response file

Reply: Thanks for the reviewer's suggestion, the cytotoxic activity assay results have been added in Supplementary Material. It can be seen from the figure that the IC50 value can’t be fitted by the dose-response curves.

Reviewer 2 Report

I understand the difficulty in doing other studies based on the amount of pure compounds obtained in this work.

Author Response

Thank you for your understanding. In subsequent experiments, we will try to find and enrich compounds 1 and 2, and then screen their other activity.